# Factors associated with attendance at screening for breast cancer: a systematic review and meta-analysis

Rebecca Mottram,[1] Wendy Lynn Knerr ![ORCID],[1] Daniel Gallacher ![ORCID],[1] Hannah Fraser,[1] Lena Al-Khudairy,[1] Abimbola Ayorinde ![ORCID],[1] Sian Williamson,[1] Chidozie Nduka,[1] Olalekan A Uthman ![ORCID],[1] Samantha Johnson,[2] Alexander Tsertsvadze,[3] Christopher Stinton,[1] Sian Taylor-Phillips ![ORCID],[1] Aileen Clarke[1]

[1]Division of Health Sciences, Warwick Medical School, University of Warwick, Coventry, UK
[2]University of Warwick Library, University of Warwick, Coventry, UK
[3]School of Epidemiology and Public Health, University of Ottawa Faculty of Medicine, Ottawa, Ontario, Canada

**Correspondence to**
Aileen Clarke;
aileen.clarke@warwick.ac.uk

## ABSTRACT

**Objective** Attendance at population-based breast cancer (mammographic) screening varies. This comprehensive systematic review and meta-analysis assesses all identified patient-level factors associated with routine population breast screening attendance.

**Design** CINAHL, Cochrane Library, Embase, Medline, OVID, PsycINFO and Web of Science were searched for studies of any design, published January 1987–June 2019, and reporting attendance in relation to at least one patient-level factor.

**Data synthesis** Independent reviewers performed screening, data extraction and quality appraisal. OR and 95% CIs were calculated for attendance for each factor and random-effects meta-analysis was undertaken where possible.

**Results** Of 19 776 studies, 335 were assessed at full text and 66 studies (n=22 150 922) were included. Risk of bias was generally low. In meta-analysis, increased attendance was associated with higher socioeconomic status (SES) (n=11 studies; OR 1.45, 95% CI: 1.20 to 1.75); higher income (n=5 studies; OR 1.96, 95% CI: 1.68 to 2.29); home ownership (n=3 studies; OR 2.16, 95% CI: 2.08 to 2.23); being non-immigrant (n=7 studies; OR 2.23, 95% CI: 2.00 to 2.48); being married/cohabiting (n=7 studies; OR 1.86, 95% CI: 1.58 to 2.19) and medium (vs low) level of education (n=6 studies; OR 1.24, 95% CI: 1.09 to 1.41). Women with previous false-positive results were less likely to reattend (n=6 studies; OR 0.77, 95% CI: 0.68 to 0.88). There were no differences by age group or by rural versus urban residence.

**Conclusions** Attendance was lower in women with lower SES, those who were immigrants, non-homeowners and those with previous false-positive results. Variations in service delivery, screening programmes and study populations may influence findings. Our findings are of univariable associations. Underlying causes of lower uptake such as practical, physical, psychological or financial barriers should be investigated.

**Trial registration number** CRD42016051597.

## STRENGTHS AND LIMITATIONS OF THIS STUDY

⇒ Comprehensive systematic review of all identified patient-level factors associated with attendance at routine population-based breast cancer (mammographic) screening.

⇒ Two reviewers independently conducted all study selection, data extraction and quality appraisal using Quality in Prognosis Studies.

⇒ Both observational and experimental designs were included, using control arms of quasi-experimental or randomised designs and ORs were independently recalculated using each study's raw data.

⇒ Heterogeneity is high partly due to the large size of studies. Studies were separately meta-analysed by study design, and sensitivity analysis was conducted for one study with an extreme effect size.

⇒ Reporting of potential confounders and effect modifiers was highly variable in studies; this was partially mitigated by recategorising variables, such as education levels, to harmonise variables across studies where possible.

## INTRODUCTION

Breast cancer was the most commonly diagnosed cancer worldwide in 2020, with 2.3 million cases, and the most common cause of cancer death in women.[1] Breast cancer incidence is higher in more developed countries (Europe, Australia, New Zealand and North America; 55.9 cases per 100 000 population) than in less developed countries (29.7 per 100 000), while the reverse is true of death rates (12.4 vs 15.0 per 100 000, respectively).[1] In the EU, mortality rates decreased 18.7% between the period 2005–2009 and 2019 from 16.44 to (predicted) 13.36 per 100 000.[2]

Population-based mammographic screening aims to reduce breast cancer mortality. However, there has been controversy about the balance of benefits and harms of breast screening[3] and breast screening programmes have become more aware of the need for promoting informed choice.[4 5]

Attendance at breast screening is not uniform among the eligible population.[6] Ross *et al*[7] described attendance at screening

as an individual decision (behavioural) which is affected by accessibility of services (structural) and by a woman's immediate surroundings (societal). Characteristics that have been associated with screening attendance can be grouped into a number of categories related to sociodemographic factors; health status; health behaviours; accessibility and logistics; beliefs, attitudes and knowledge; simple intention to attend and societal factors including health systems financing and organisation.[8–11]

Most reviews of factors associated with breast screening attendance have focused on individual factors.[12–14] We aimed to provide a comprehensive systematic review of all identified patient-level characteristics associated with the uptake of population-based mammographic screening, to inform screening programmes of the available evidence about who does and does not attend.

## METHODS

### Protocol and registration

The review was conducted in accordance with prespecified methods documented in the protocol registered on the 22November 2016 in the PROSPERO International Prospective Register of Systematic Reviews database (online supplemental file A).[15]

### Search and information sources

The Cumulative Index to Nursing and Allied Health Literature (CINAHL), Cochrane Library, Embase, Medline, PsycINFO and Web of Science were searched for studies published between 1 January 1987 and 26 June 2019. The search was developed in Medline using a combination of MeSH headings and free-text terms and adapted for use in the other databases (the search strategy is available in online supplemental file B).

Reference lists of relevant reviews were searched for potentially relevant studies. Experienced researchers with prior studies in the field were contacted to identify other potentially relevant studies that had not been identified in the searches.

### Eligibility criteria

Primary studies of any design were included if they reported attendance data from routine population-based mammography screening programmes in relation to at least one patient-level factor, and were written in English between January 1987 and June 2019. Studies were excluded if they involved self-reported mammography uptake, opportunistic screening programmes, data for only a subgroup of the eligible population (eg, only women in a narrow age range, only immigrants or only rural women) or uptake data by number of invitations sent rather than number of women. Reviews, commentaries, opinions, letters, and non-empirical and qualitative studies were excluded.

### Study selection and data extraction process

Pairs of reviewers screened titles and abstracts independently to identify potentially relevant studies with

third reviewer cross-check. Two reviewers independently assessed full-text studies for formal inclusion/exclusion assessment against predefined eligibility criteria with third reviewer cross-check. Disagreements were resolved by a consensus between the two reviewers or by help of a third reviewer.

Data from included studies were extracted and then cross-checked by two reviewers independently. The data included the number of women who attended mammographic screening and the number invited, and data on patient characteristics, including: sociodemographic factors, such as age, marital status, educational level, race/ethnicity, immigration status and socioeconomic status (SES, which was measured in two ways, (a) with various composite indices of deprivation that included factors such as housing density, employment, education, social support, car ownership and crime prevalence, and (b) based on household income); beliefs, attitudes and socioemotional factors; health history and behaviours; logistic and accessibility factors (eg, distance from screening centre).

### Risk of bias of included studies

Risk of bias (RoB) of all included studies was appraised by two independent reviewers using the Quality in Prognosis Studies (QUIPS) tool.[16] The QUIPS tool covers six RoB domains (participation, attrition, prognostic factor, confounding factors, outcome measurement and analysis and reporting), each of which includes multiple items that are judged separately. A conclusive judgement for each RoB domain is reached and expressed on a three-grade scale (high, moderate or low RoB).

### Synthesis of data

We used raw attendance data to calculate unadjusted ORs for each factor. A random-effects model-based meta-analysis was conducted for an association between a factor of interest (dichotomous or more categories) and the dichotomous outcome (screening attendance) to generate Mantel-Haenszel ORs with 95% CIs, when possible.[17] Random-effects models were used to allow for heterogeneity in the effects of the factors considered to vary across the different studies.

In addition to the main meta-analyses, we conducted separate meta-analyses for (a) observational studies whose samples were made up only of women who had previously attended screening (hereafter referred to as rescreening studies) and (b) intervention studies (quasi-experimental and randomised controlled trials) that reported characteristics separately for intervention and control arms, recording only data for the control group, as their attendance would not be influenced by exposure to an intervention. We also conducted a sensitivity analysis to determine the impact of a study with an extreme effect size[18] on the meta-analysis of SES.

We summarised results narratively if there were inadequate quantitative data for meta-analysis, if variables were reported in fewer than three studies,[17] or if the data from

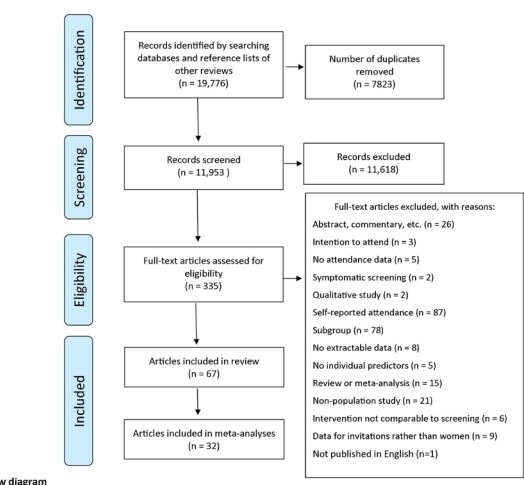

**Figure 1. PRISMA flow diagram**

**Figure 1** PRISMA flow diagram, showing the process of study flow and reasons for exclusion. The searches of electronic databases identified 11 953 unique publications (after deduplication), published between January 1987 and June 2019, of which 11 618 were excluded at the level of abstract/title screening, leaving 335 records for full-text review. Of the 335 full texts, 66 unique studies reported in 67 publications were included. PRISMA, Preferred Reporting Items for Systematic Reviews and Meta-Analyses.

multiple studies were highly variable and therefore could not be meaningfully pooled.

This review is reported according to Preferred Reporting Items for Systematic Reviews and Meta-Analyses guidelines (online supplemental file C).[19] All analyses were conducted in Stata V.16.

### Patient and public involvement

Public contributors were involved in design and informed of ongoing progress and findings as part of the West Midlands Centres for Leadership in Applied Health Research. Results were reported back to the contributors as part of the wider dissemination activities of the relevant theme in the Centres for Leadership in Applied Health Research.

## RESULTS
### Literature search

The process of study flow and reasons for exclusion are provided in figure 1. In brief, the searches of electronic databases identified 11 953 unique publications (after deduplication), published between January 1987 and June 2019, of which 11 618 were excluded at the level of abstract/title screening, leaving 335 records for full-text review. Of the 335 full texts, 66 unique studies reported in 67 publications were included.[18 20–87]

### Study characteristics

Characteristics of all included studies are listed in online supplemental file D. Of the 66 studies, 49 were observational (45 retrospective cohort, 2 cross-sectional and 2 case–control designs); and 17 were intervention studies (16 randomised controlled trials and 1 quasi-experimental). Sample sizes ranged from 82 to 4.8 million.

The studies were conducted in Europe (n=40), North America (n=18), Asia-Pacific (n=5) and the Middle East (n=3). The UK had the most studies (n=16) followed by the USA (n=11).

We were able to pool data from 31 observational studies (reported in 32 publications) on the attendance at screening in relation to nine factors (age, education, home ownership, immigration status, marital status, results of previous mammogram, rural/urban residence, SES and income) (table 1). We were only able to pool data from three intervention studies, and only for one factor (age).

Adequate data for meta-analysis was not provided for 35 studies; although six of these studies provided adequate data to calculate ORs and CIs, and are narratively reported in table 2. The remaining 29 studies reported data that could not be analysed. (Reasons are detailed in online supplemental file E.) In brief, 14 of the 29 studies were intervention trials, where data were not in the right format for us to use. The other 15 studies could not be analysed because uptake data were reported by health-provider characteristics rather than patient characteristics; because the paper reported percentage uptake but not sample sizes per category; or because data for different factors were not reported separately.

### Risk of bias

RoB across studies was generally low on all domains (figure 2). For study participation, 71% of studies were considered at low RoB; for attrition, 91%; for outcome measurement, 97% and for statistical analysis and reporting, 83%. For measurement of variables associated with attendance (prognostic factors), more than half (61%) of studies had a low RoB, while 23% had a high RoB, mostly due to SES being measured at the area level (eg, neighbourhood) rather than at the individual level. More than half of studies (53%) had a low RoB with regard to measuring potential confounders, with around one-quarter (27%) having a moderate risk and just over one-fifth (21%) having a high risk.

### Quantitative data analysis (meta-analyses)

Table 1 presents unadjusted OR estimates with their 95% CIs of attendance at breast screening for factors that were reported in three or more studies. The analyses gave $I^2$ values of around 99%, meaning that there was a high level of heterogeneity, except for the analysis of homeowners versus tenants, where the $I^2$ value was 38.9% (table 1).

We compared the odds of attending mammographic screening by the age bands most commonly eligible for national screening programmes (60—69 and 50—59). There was no significant difference by age group in meta-analyses of observational studies (n=16; OR 0.97, 95% CI:

**Table 1**  Results of meta-analyses*

| Variables | Number of women (number of studies included)† | % uptake | OR of attendance (unadjusted): range \| overall (95% CI) |
|---|---|---|---|
| Age (60—69 vs 50—59)‡ | | | |
| Observational studies | 5 065 779 (16) | 56 vs 55 | 0.65 to 1.42 \| 0.97 (0.88 to 1.08) |
| Intervention studies | 2343 (3) | 52 vs 57 | 0.24 to 1.16 \| 0.78 (0.47 to 1.31) |
| Rescreening studies (age at initial screen) | 271 641 (3) | 74 vs 74 | 0.93 to 1.05 \| 0.99 (0.93 to 1.06) |
| Education level | 550 646 (6) | | |
| Medium vs low | | 83 vs 77 | 1.05 to 1.45 \| 1.24 (1.09 to 1.41) |
| High vs low | | 81 vs 77 | 0.76 to 1.31 \| 1.10 (0.97 to 1.26) |
| High vs medium | | 81 vs 83 | 0.61 to 1.10 \| 0.89 (0.78 to 1.02) |
| Housing tenure (homeowner vs tenant/non-owner) | 223 293 (3) | 84 vs 70 | 2.06 to 2.20 \| 2.16 (2.08 to 2.23) |
| Country of origin (non-immigrants vs immigrants) | 2 409 902 (7) | 81 vs 60 | 1.75 to 2.81 \| 2.23 (2.00 to 2.48) |
| Income | 1 193 238 (5) | | |
| Intermediate vs low | | 77 vs 66 | 1.78 to 2.09 \| 1.96 (1.68 to 2.29) |
| High vs low | | 80 vs 66 | 1.61 to 2.87 \| 2.18 (1.86 to 2.56) |
| High vs intermediate | | 80 vs 77 | 0.81 to 1.37 \| 1.11 (0.95 to 1.30) |
| Marital status | 1 293 753 (7) | 80 vs 69 | 1.38 to 2.36 \| 1.86 (1.58 to 2.19) |
| (Married/cohabiting vs unmarried/non-cohabiting) | | | |
| Residence (rural vs urban) | 65 641(3) | 74 vs 65 | 0.80 to 1.59 \| 1.12 (0.76 to 1.66) |
| Previous result of mammogram (rescreening studies only: false positive vs normal) | 3 540 953 (6) | 60 vs 68 | 0.49 to 0.89 \| 0.77 (0.68 to 0.88) |
| Socioeconomic status (SES) | 6 600 283 (11) | | |
| Medium vs low | | 56 vs 48 | 1.08 to 2.35 \| 1.45 (1.20 to 1.75) |
| High vs low | | 54 vs 48 | 0.75 to 3.59 \| 1.69 (1.40 to 2.05)§ |
| High vs medium | | 54 vs 56 | 0.69 to 1.53 \| 1.17 (0.96 to 1.41) |

*All results in this table are for observational studies except the data for age, which includes results for the separate meta-analysis of intervention studies.

†References for studies pooled for meta-analyses of observational studies are provided in forest plots in figures 3 and 4.

‡We focused on the age bands most commonly eligible in population-based programmes and did not analyse odds for those younger than age 50 or older than 69.

§The ORs and CIs for SES include all relevant observational studies. We also performed a sensitivity analysis by removing the large study from France by DeBorde et al,[18] which found that women with high or medium SES were both more likely to attend compared with women of lower SES (OR 1.84, 95% CI: 1.55 to 2.17, p<0.001; and OR 1.49, 95% CI: 1.27 to 1.76, p<0.001, respectively).

0.88 to 1.08, p=0.631, figure 3) or intervention trials (n=3; OR 0.78, 95% CI: 0.47 to 1.31, p=0.354).

We grouped education data from six studies to approximate the United Nations Educational, Scientific and Cultural Organisation (UNESCO) three-level classification: low (≤10 years), middle (11–15 years) and high (>15 years). Compared with women with a low level of education, women with a medium level were more likely to attend (OR 1.24, 95% CI: 1.09 to 1.41, p<0.001). Results from comparisons of women with a high level of education versus low or medium levels were not statistically significant (figure 4A).

The odds of attending mammographic screening were higher for homeowners than for tenants or non-owners (n=3; OR 2.16, 95% CI: 2.08 to 2.23, p<0.001, figure 3).

Meta-analysis of participants' country of origin showed that people born in the study country (non-immigrants) were more likely to attend than immigrants (n=7; OR 2.23, 95% CI: 2.00 to 2.48, p<0.001, figure 3).

We meta-analysed attendance using two measures of SES. Data for overall SES from 11 studies were grouped into low, medium and high categories. Women with medium or high SES were more likely to attend than those with a low SES (medium vs low SES OR 1.45, 95% CI: 1.20 to 1.75, p<0.001; high vs low SES OR 1.69, 95% CI: 1.40 to 2.05, p<0.001, figure 4B). One study from France (DeBorde)[18] (n=4.8 million) reported that women with a higher SES were less likely to attend than those with either a low or intermediate SES. We conducted a sensitivity analysis excluding that study, but it made very little

**Table 2** Likelihood of attending screening by factors not suitable for meta-analysis in observational studies

| Variable | N* | Included studies | % uptake: variable vs reference category | OR (95% CI) |
|---|---|---|---|---|
| Less likely to attend | | | | |
| No access to vehicle | 144 181 | Jensen 2012b | 61 vs 82 | 0.33 (0.32 to 0.34) |
| | 37 059 | O'Reilly 2012 | 60 vs 78 | 0.43 (0.41 to 0.46) |
| Negative attitude about breast screening | 497 | Kee 1993 | 53 vs 60 | 0.44 (0.35 to 0.55) |
| Receiving disability benefits | 885 979 | Le 2019 | 69 vs 76 | 0.70 (0.70 to 0.71) |
| First invitation to screening | 742 786 | Renshaw 2010 | 40 vs 76 | 0.22 (0.21 to 0.22) |
| Spoken/preferred language not English | 18 851 | Blanchard 2004 | 62 vs 83 | 0.33 (0.28 to 0.39) |
| | 43 819 | Tatla 2003 | 60 vs 78 | 0.43 (0.41 to 0.46) |
| Long-term limiting illness | 37 059 | O'Reilly 2012 | 71 vs 77 | 0.71 (0.68 to 0.75) |
| | 144 264 | Jensen 2015b | 71 vs 80 | 0.64 (0.61 to 0.66) |
| Smoking (current) | 28 874 | Katz 2018 | 84 vs 88 | 0.72 (0.65 to 0.79) |
| Living in crowded housing conditions | 31 948 | Zackrisson 2004 | 37 vs 66 | 0.29 (0.24 to 0.36) |
| Employment status | | | | |
| Outside workforce vs employed/self-employed | 640 843 | Le 2019 | 63 vs 77 | 0.51 (0.50 to 0.51) |
| | 119 269 | Jensen 2012b | 77 vs 83 | 0.66 (0.64 to 0.68) |
| Unemployed vs employed/self-employed | 481 911 | Le 2019 | 61 vs 77 | 0.47 (0.45 to 0.49) |
| | 102 178 | Jensen 2012b | 67 vs 83 | 0.41 (0.40 to 0.43) |
| Number of childbirths | 46 041 | Lagerlund 2002 | | |
| 0 vs 1–2 | | | 82 vs 91 | 0.44 (0.40 to 0.48) |
| 3+ vs 1–2 | | | 90 vs 91 | 0.81 (0.75 to 0.87) |
| No family history of BC | 119 502 | O'Byrne 2000 | 85 vs 86 | 0.90 (0.86 to 0.94) |
| Type of clinic (mobile vs fixed) | 119 502 | O'Byrne 2000 | 84 vs 85 | 0.93 (0.88 to 0.98) |
| Schizophrenia | 110 240 | Chochinov 2009 | 45 vs 58 | 0.58 (0.52 to 0.64) |
| More likely to attend | | | | |
| No comorbidities | 76 520 | Larsen 2018 | 82 vs 75 | 1.53 (1.46 to 1.60) |
| 60+ primary care visits during 6-year study period (vs <60) | 43 968 | Katz 2018 | 91 vs 79 | 2.70 (2.55 to 2.86) |
| Depression | 38 823 | Katz 2018 | 86 vs 85 | 1.12 (1.02 to 1.23) |
| Good general health | 37 059 | O'Reilly 2012 | 77 vs 68 | 1.55 (1.46 to 1.64) |
| Heart disease | 6501 | Katz 2018 | 90 vs 85 | 1.75 (1.61 to 1.91) |
| Not living in capital city | 885 979 | Le 2019 | 76 vs 62 | 1.94 (1.91 to 1.97) |
| Previous attender | 11 664 | Taylor-Phillips 2013 | 73 vs 45 | 3.32 (3.05 to 3.61) |
| Citizen of country | 885 979 | Le 2019 | 75 vs 51 | 2.88 (2.82 to 2.94) |
| Member of majority racial/ethnic group | 17 997 | Blanchard 2004 | 85 vs 75 | 1.70 (1.52 to 1.89) |
| Religion | | | | |
| Catholic vs none | 37 140 | O'Reilly 2012 | 74 vs 68 | 1.40 (1.25 to 1.47) |
| Protestant vs none | | O'Reilly 2012 | 77 vs 68 | 1.57 (1.46 to 1.70) |
| Never HRT use | 119 502 | O'Byrne 2000 | 16 vs 14 | 1.13 (1.09 to 1.17) |
| Referral by health professional | 56 420 | Tatla 2003 | 77 vs 76 | 1.05 (1.00 to 1.10) |
| No difference in attendance or mixed results | | | | |
| BMI | 19 168 | Katz 2018 | 87 vs 87 | 0.95 (0.87 to 1.04) |
| >0 GPs per 100 000 inhabitants | 4865 | Pornet 2010 | 55 vs 56 | 0.96 (0.85 to 1.08) |
| >0 radiologists per 100 000 inhabitants | 4865 | Pornet 2010 | 52 vs 56 | 0.87 (0.72 to 1.05) |

**Table 2**  Continued

| Variable | N* | Included studies | % uptake: variable vs reference category | OR (95% CI) |
|---|---|---|---|---|
| Diabetes | 9849 | Katz 2018 | 87 vs 84 | 1.25 (1.17 to 1.33) |
| | 504 288 | Chan 2014 | 60 vs 66 | 0.79 (0.78 to 0.80) |
| Distance to screening centre | 137 419 | Jensen 2012b | 77 vs 80 | 0.86 (0.84 to 0.88) |
| | 833 856 | St-Jacques 2013 | 53 vs 52 | 1.02 (1.01 to 1.03) |
| | 13 260 | Ouédraogo 2014 | 54 vs 50 | 0.85 (0.79 to 0.91) |
| Physician years since graduation | 105 575 | Makedonov 2015 | 74 vs 75 | 1.03 (0.99 to 1.06) |

*Reflects the number of participants analysed for each factor, which can differ for different factors in the same study depending on data availability.

BMI, body mass index; GPs, general practitioners; HRT, hormone replacement therapy.

difference to the odds of attending: women with high or medium SES were both more likely to attend compared with women of lower SES (OR 1.84, 95% CI: 1.55 to 2.17, p<0.001, and OR 1.49, 95% CI: 1.27 to 1.76, respectively).

Data on income from five studies were grouped into low, intermediate and high categories. Women with an intermediate or high income were more likely to attend than those with low income (intermediate vs low income OR 1.96, 95% CI: 1.68 to 2.29, p<0.001; high vs low OR 2.18, 95% CI: 1.86 to 2.56, p<0.001; high vs intermediate OR 1.11, 95% CI: 0.95 to 1.30, p=0.20, figure 4C). For both income and SES, there was no significant difference between women at intermediate and high levels, indicating that there was no statistically significant dose response effect for higher SES or income.

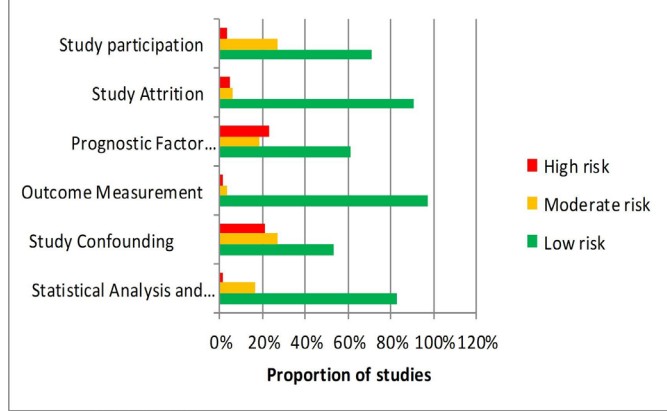

**Figure 2**  Overall summary of QUIPS risk of bias scores: risk of bias (RoB) of all included studies was appraised by two independent reviewers using the Quality in Prognosis Studies (QUIPS) tool. The QUIPS tool covers six RoB domains (participation, attrition, prognostic factor, confounding factors, outcome measurement and analysis and reporting), each of which includes multiple items that are judged separately. A conclusive judgement for each RoB domain is reached and expressed on a three-grade scale (high, moderate or low RoB). RoB across studies was generally low on all domains.

Women who were married or cohabiting were more likely to attend than their unmarried or non-cohabiting counterparts (n=7; OR 1.86, 95% CI: 1.58 to 2.19, p<0.001, figure 3).

We analysed data separately for studies with samples made up only of women who had previously attended mammographic screening (ie, rescreening studies). Six of these studies reported data on attendance based on the results of a previous mammogram. Women who had previously received a false-positive result were less likely to attend than those with a normal result (OR 0.78, 95% CI: 0.68 to 0.88, p<0.001, figure 3).

There was no statistically significant difference in attendance among women living in rural compared with urban areas (n=3; OR 1.12, 95% CI: 0.76 to 1.66, p=0.557).

### Narrative synthesis

Factors that could not be meta-analysed (because they were reported in fewer than three studies or could not be pooled) are reported in table 2 with ORs.

These studies include a variety of factors associated with reduced attendance clustered around sociodemographic, accessibility and logistics (living in crowded housing and being unemployed, receiving disability benefits, lack of access to a vehicle), and spoken language not English.

Associations with women's health status, behaviours, attitudes and knowledge showed a mixed picture. There was some evidence that good general health, lack of comorbidity and not taking hormone replacement therapy (HRT) were all associated with higher attendance, but studies also reported higher attendance among women with a higher numbers of previous clinic visits, depression and heart disease. A previous negative attitude to breast screening, limiting long-term illness, schizophrenia, non-work-related stress and current smoking were associated with lower attendance.

Factors that did not show any statistical difference included body mass index and service provision factors. No difference in women's attendance was found according to availability of general practitioners or radiologists or physician years since graduation, and there were

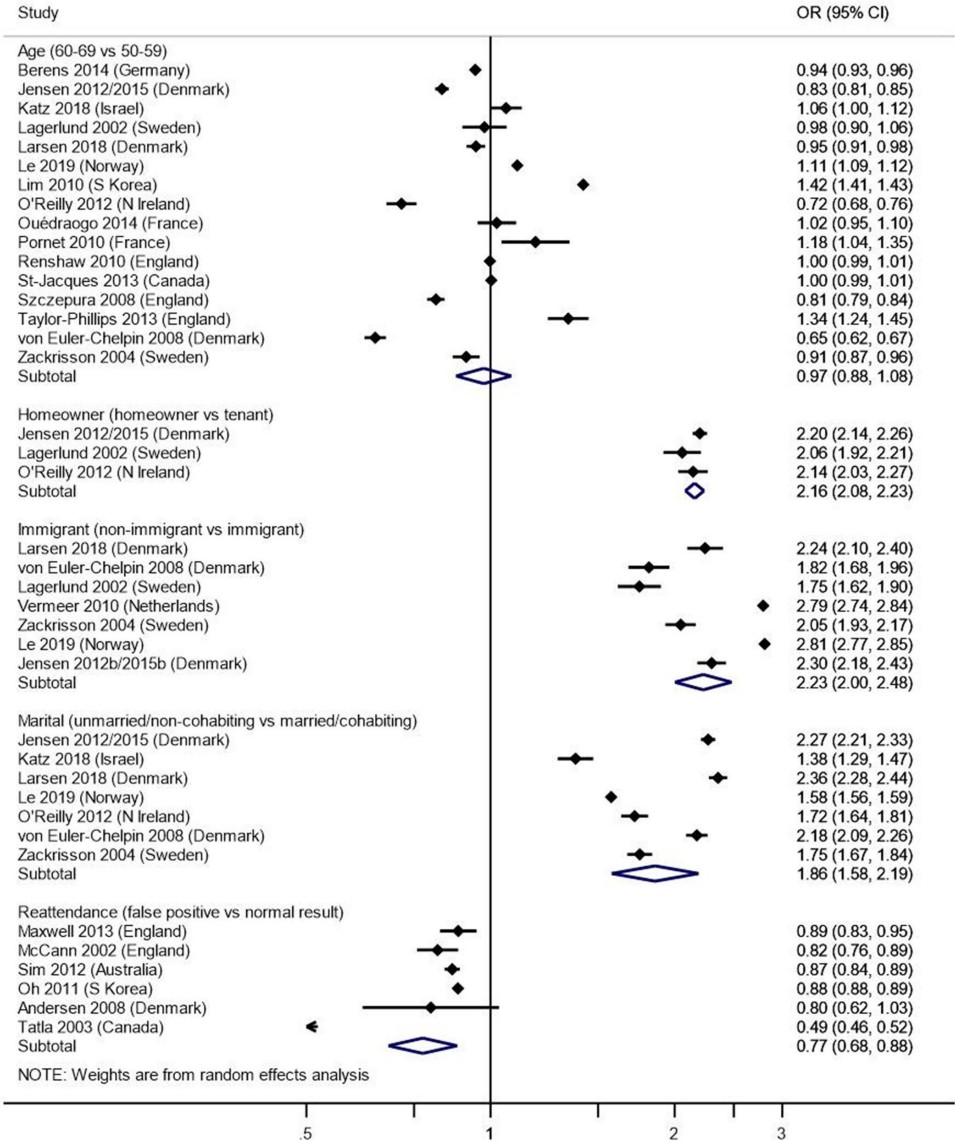

**Figure 3** Meta-analyses. This figure shows comparisons of the odds of attending mammographic screening, using random-effects analysis, in observational studies by the following variables. Points to the left of the centre line (<1) suggest a lower likelihood of attending screening, while points to the right of the centre line (>1) indicate a higher likelihood of attending. Age bands: we compared the age bands most commonly eligible for national screening programmes (60—69 and 50—59); there was no significant difference by age group (n=16; OR 0.97, 95% CI: 0.88 to 1.08, p=0.631); Home ownership: we compared people who own their homes to those who are tenants or do not own their homes; the odds of attending were higher for homeowners than for tenants or non-owners (n=3; OR 2.16, 95% CI: 2.08 to 2.23, p<0.001); Immigrant status: we compared screening attendance of people born in the country in which the study took place (non-immigrants) to those born in another country (immigrants); non-immigrants were more likely to attend than immigrants (n=7; OR 2.23, 95% CI: 2.00 to 2.48, p<0.001). Marital status: we compared women who were married or cohabiting to those who were unmarried or not cohabiting: women where were married/cohabiting were more likely to attend than their unmarried/non-cohabiting counterparts (n=7; OR 1.86, 95% CI: 1.58 to 2.19, p<0.001). Reattendance; using data from studies with samples made up only of women who had previously attended mammographic screening, we compared women who had previously received a false-positive to those who had had a normal result; those with a previous false-positive result were less likely to reattend (OR 0.78, 95% CI: 0.68 to 0.88, p<0.001).

mixed results according to distance to screening centre and diabetes.

## DISCUSSION

We undertook a comprehensive review of the current evidence on patient-level factors associated with breast cancer (mammographic) screening attendance. Where appropriate, meta-analyses were performed to determine the strength of association.

### Main findings

In line with other systematic reviews, we found that in general higher SES status, higher income,[14] being born in the country of residence (ie, non-immigrant)[12] and home ownership (compared with renting) predicted

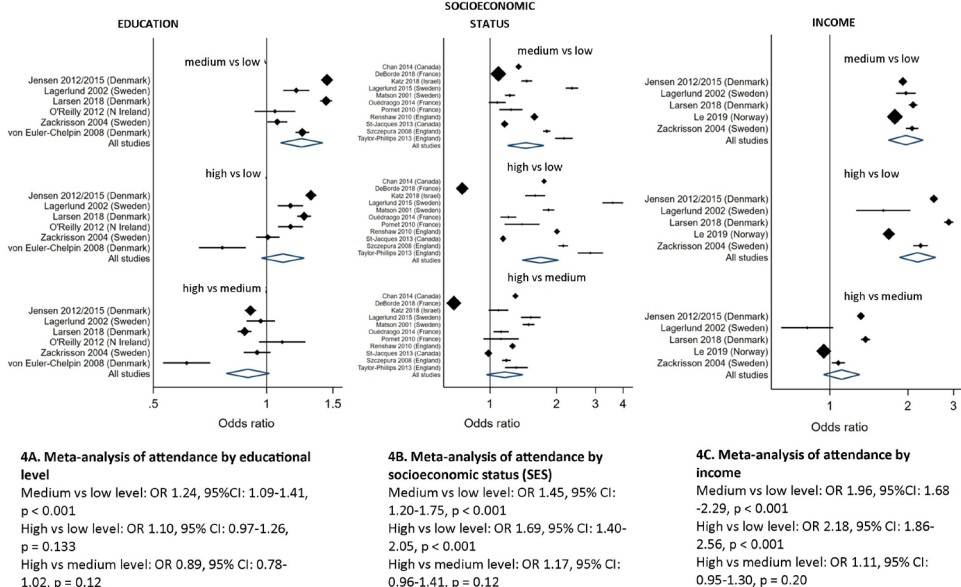

**Figure 4** Meta-analyses of attendance by educational level, socioeconomic status (SES) and income. These figures show random-effects meta-analyses of screening attendance by educational level and socioeconomic status in observational studies. Points to the left of the centre line (<1) suggest a lower likelihood of attending screening, while points to the right of the centre line (>1) indicate a higher likelihood of attending. Figure 4A shows the effects of different levels of education on screening attendance. We grouped education data to approximate the United Nations Educational, Scientific and Cultural Organization (UNESCO) three-level classification: low (≤10 years), middle (11–15 years) and high (>15 years). Compared with women with a low level of education, women with a medium level were more likely to attend (OR 1.24, 95% CI: 1.09 to 1.41, p<0.001). Results from comparisons of women with a high level of education versus low or medium levels were not statistically significant (figure 4A). Figure 4B shows the meta-analysis of attendance by overall SES. Studies were grouped into low, medium and high categories. Women with medium or high SES were more likely to attend than those with a low SES (medium vs low SES OR 1.45, 95% CI: 1.20 to 1.75, p<0.001; high vs low SES OR 1.69, 95% CI: 1.40 to 2.05, p<0.001, figure 4B). Figure 4C shows the meta-analysis of screening attendance by income. Studies were grouped into low, intermediate and high categories. Women with an intermediate or high income were more likely to attend than those with low income (intermediate vs low income OR 1.96, 95% CI: 1.68 to 2.29, p<0.001; high vs low OR 2.18, 95% CI: 1.86 to 2.56, p<0.001; high vs intermediate OR 1.11, 95% CI: 0.95 to 1.30, p=0.20, figure 4C). For both income and SES, there was no significant difference between women at intermediate and high levels, indicating that there was no statistically significant dose response effect for higher SES or income.

mammographic screening attendance. However, it appears that women with a higher SES or income were not more likely to attend than those with an intermediate level. We hypothesise that women with a higher SES may be more likely to use alternative screening services (ie, opportunistic or privately funded screening) compared with women with a low or intermediate SES, thus their attendance would not be apparent in studies using data from national screening programmes. This was suggested as a limitation by many of the included studies in this review, most notably the large study from France[18] (n=4.8 million), which was the only study to find that women with a higher SES were less likely to attend than those with either a low or intermediate SES. The authors of that study note the high levels of opportunistic screening available to women with a high SES in France. We conducted a sensitivity analysis excluding that study, but it made very little difference to the ORs for attendance.

A medium level of education was also associated with screening attendance when compared with a low level, but a higher level of education was not associated with increased attendance compared with either medium or lower levels. As with the analyses of SES, it is possible that

women with the highest levels of education are more likely to use alternative screening services not reflected in data from public screening programmes.

We hypothesised that some variation in relation to education or SES might be due to changes in women's attitudes to breast screening as a result of concerns about its overall benefits,[65 88] perhaps related to the informed-choice agenda.[4] However, we found no population screening studies investigating this.

Our results also support previous research indicating that marital status is associated with attendance at mammography,[65 88–91] with women who were married or cohabiting more likely to attend than their unmarried or non-cohabiting counterparts. Previous literature indicates lower uptake among women from minority-ethnic backgrounds.[92 93] While our data were not sufficient to meta-analyse ethnicity, we did find that immigrant women were less likely to attend screening than non-immigrants.

We did not find a significant effect of age. There was very high heterogeneity here, with individual large studies finding highly statistically significant results in both directions. We hypothesised that attendance may be higher among older women because they have been invited

to breast screening for at least two decades, and attendance may have become more routine in this cohort, and possibly less likely to be affected by recent debates around the risks and benefits of screening. To explore this, we did a post-hoc analysis of the effect of age on attendance by the year of study completion. We found that older women were more likely to attend compared with younger women in more recent studies (ie, those completed since 2010), but that the opposite was true in older studies, particularly those published before 2005.

Women who received a false-positive result at a previous screening were less likely to attend than those with a normal result, confirming previous findings.[94]

### Strengths and limitations

This review has many strengths. The large number of studies included (n=66), involving more than 22 million women, represents a comprehensive overview of available evidence. Studies included in the meta-analysis were judged to have a low RoB on most domains and included large numbers of women. At least two reviewers were involved at all stages to reduce the risk of errors and bias. This study was undertaken from the perspective of population-based breast cancer screening programmes and we were strict in our eligibility criteria in including only those studies. Studies where the sampling frame was restricted to population subgroups (and not based on population-based screening programmes) were excluded. We also excluded studies that relied on self-reported attendance (though it is important to note that self-report is essential for some factors, such as ethnicity and attitudes to screening).

A limitation is that most studies reported cross-sectional attendance data, which included mixed groups of those who were attending for the first time and some who had previously attended. Also, we inevitably had to make choices of categories for meta-analysis which may affect meta-analytic results; where possible we used independent sources to select appropriate categorisations.

The main limitation of this review is significant between-study heterogeneity. Although we used random-effect models throughout, our results should be considered in light of this. We chose random-effects models as almost all of our analyses contained heterogeneity and it is also expected that there would be differences in attendance across the different study populations. Studies with larger sample sizes are assumed to contain the least uncertainty and are given higher weightings than smaller studies. For analyses of small numbers of studies, the random-effects analysis may struggle to correctly estimate uncertainty, but any meta-analysis performed on few studies would have its limitations, and the use of random-effects analysis maintained consistency with the other analyses.

Heterogeneity may in part be due to differences between health systems and the organisation of mammographic screening, as well as differences in the culture and attitudes of the populations served. We conducted sensitivity analysis to determine the impact of a very large study with an extreme effect size[18] on the meta-analysis of SES. For some outcomes (such as age), the heterogeneity encompasses studies with highly significant results in both directions, and here the results of the meta-analysis should be interpreted with great caution. For other variables (such as reattendance after false-positive results), the high $I^2$ simply reflects that there were very large studies with very small CIs, which all had point estimates of different magnitude in the same direction. Here the meta-analysis results show a consistent effect, with some disagreement between studies on the exact size of effect.

Another limitation is that we extracted univariable associations with uptake. In practice, many of the variables investigated will be highly correlated, and there will be complex interactions and confounding which we have not been able to account for. While some studies did report multivariable models, these were varied in structure, methods and variables included, so would have been difficult to combine in any meaningful way. We were therefore unable to undertake multivariable meta-regression analysis, examining the effects of individual attendance factors on overall attendance.

For the studies included in the narrative analysis, large numbers of women were also often involved, but these studies should be treated with caution as they are potentially subject to bias. The risk of confounding was found to be high in these studies using the QUIPs tool. However, confounding is inherent in the design of population-based observational and especially ecological designs.

To investigate the risk of reporting bias, we conducted funnel plots (online supplemental file F), which demonstrated the high level of heterogeneity present between the studies in our analyses. Age was the only analysis where the studies disagree over the direction of attendance, however the disagreement is among larger studies, suggesting this is unlikely to be associated with biased reporting and instead down to the study heterogeneity. All other analyses, while having studies which disagree on the point estimate, have agreement as to which group is more or less likely to attend mammographic screening. Overall, we are not concerned about reporting bias.

Finally, we have not included health insurance (or lack of health insurance) as a factor in the narrative analysis because of the problems of comparison between countries.

## CONCLUSIONS

A wide variety of factors affect a woman's decision to attend breast screening. Our main findings are that attendance was lower in women with lower SES, those who were immigrants, non-homeowners and those with previous false-positive results. Based on our current findings, if screening programmes wish to improve equity of access to breast screening services, they should concentrate on women facing access (practical, physical, psychological and financial) barriers.

Future research in this area would also need to systematically assess the effects of interventions to reduce the impact of access barriers to screening attendance.

### Deviations from study protocol

To assess RoB, the QUIPS tool was used rather than the Quality Assessment Tool; and for data synthesis, despite significant heterogeneity, meta-analysis was possible for some predictors. In addition, we clarified our inclusion criteria to include only studies with data from routine population-based mammography screening programmes in order to ensure generalisability.

**Acknowledgements** The authors would like to thank Magdalena Skrybant, Patient and Public Involvement and Engagement Lead, Applied Research Centre West Midlands (formerly Centres for Leadership in Applied Health and Care, West Midlands) for her support in coordinating public engagement in this project.

**Contributors** RM conceived the study as part of her PhD Dissertation, and it was further refined in collaboration with AC and ST-P. AC, CS and WLK further developed the inclusion and exclusion criteria. SJ undertook database searches and AC, HF, RM, LA-K, SW and WLK reviewed titles and abstracts. Each study retained for full-text review was reviewed by RM and WLK. Discrepancies regarding inclusion and exclusion were resolved by AC and CS. RM and WLK did data extraction, and data were checked by OAU and CN. Studies were critically appraised by AA, AT, CS and WLK. Meta-analyses were conducted by DG. Thematic synthesis was done by AC, ST-P and WLK. All authors contributed to the manuscript and approved the final version. AC is the guarantor for this paper.

**Funding** This research was funded by the National Institute for Health Research Collaboration for Leadership in Applied Health Research and Care West Midlands (NIHR CLAHRC WM), now recommissioned as NIHR Applied Research Collaboration West Midlands (NIHR ARC WM). RM, L A-K, ST-P, SW, HF, CS and AC were all supported by the NIHR CLAHRC WM and WLK, AA, AC and L-AK are all partly supported by the NIHR ARC WM. OAU is supported by the NIHR using Official Development Assistance (ODA) funding. ST-P is supported by an NIHR Career Development Fellowship (CDF-2016-09-018). The views expressed in this publication are those of the author(s) and not necessarily those of the UK National Health Service (NHS), the NIHR or the Department of Health and Social Care.

**Competing interests** RM reports personal fees from the National Institute for Health Research (NIHR) Centre for Leadership in Applied Research and Health Care (CLARHC) West Midlands during the conduct of the study. LA-K reports grants from the NIHR during the conduct of the study. ST-P reports grants from NIHR outside the submitted work. AC reports grants from the NIHR for the NIHR Applied Research Centre (ARC) West Midlands and previously from the NIHR CLARHC West Midlands, which supported her and researchers working on this project. AC also received grants from Public Health England (PHE) outside the submitted work.

**Patient consent for publication** Not required.

**Provenance and peer review** Not commissioned; externally peer reviewed.

**Data availability statement** No data are available. No new data have been created in the preparation of this report and therefore there is nothing available for access and further sharing. All queries should be submitted to the corresponding author.

**ORCID iDs**
Wendy Lynn Knerr http://orcid.org/0000-0002-6475-9417
Daniel Gallacher http://orcid.org/0000-0003-0506-9384
Abimbola Ayorinde http://orcid.org/0000-0002-4915-5092
Olalekan A Uthman http://orcid.org/0000-0002-8567-3081
Sian Taylor-Phillips http://orcid.org/0000-0002-1841-4346

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
