## [Reviewer comments · BMJ Open]

ARTICLE DETAILS

TITLE (PROVISIONAL)	Factors associated with attendance at screening for breast cancer: a systematic review and meta-analysis
AUTHORS	Mottram, Rebecca; Knerr, Wendy; Gallacher, Daniel; Fraser, Hannah; Al-Khudairy, Lena; Ayorinde, Abimbola; Williamson, Sian; Nduka, Chidozie; Uthman, Olalekan; Johnson, Samantha; Tsertsvadze, Alexander; Stinton, Christopher; Taylor-Phillips, Sian; Clarke, Aileen

VERSION 1 – REVIEW

REVIEWER	Amanda Dibden Queen Mary University of London, Centre for Cancer Prevention
REVIEW RETURNED	09-Dec-2020

GENERAL COMMENTS	This aim of this paper is to provide an up to date synthesis of various factors that may influence attendance to breast screening. The review is well written and thorough, and encompasses a large number of factors that have been investigated for possible associations with screening attendance. Where multiple studies assessing the same factor exists, the authors have used random effects meta-analysis to produce a pooled estimate. Their findings suggest that SES, income and educational status among others have an effect on attendance to screening. I have just a few minor points: Methods: What software was used to conduct the meta-analysis? Results: 1. The results are presented clearly although it would be helpful for the papers included in the meta analysis and narrative review to be referenced in the main body of the text rather than having to look at the supplementary material2. What scales were used to categorise women into low, medium and high income and SES groups?3. Usual convention is to present p-values as $p < 0.001$ rather than $p = 0.000$ or $p < 0.000$ Table 1. It would be useful to have a column specifying the range of the ORs for each factor Discussion: The review contains both RCT and observational study data. Have authors considered the influence of a given the study design on the results in terms of potential biases? Figures 3-10: It would be helpful if the text size was larger Supplementary file E: It would be useful to have two additional columns here – the factor being analysed (e.g. SES, income, age etc.) and the odds ratios. Supplementary file F: Lagerlund reference needs completing
--

REVIEWER	Sameer Bhargava
REVIEW RETURNED	30-Jan-2021

GENERAL COMMENTS

Dear editors and authors,
 Thank you for giving me the opportunity to review this paper about factors associated with attendance at breast cancer screening. The paper is well written and gave a good overview of factors associated with attendance, and I appreciated having the opportunity to review it. There are some issues that I would like the authors to consider in their revision. As some points are repeated in the text, my comments may apply to other places in your text than where I direct the comment.

Page 1, 17 and 18: Please refer to the ICMJE criteria for authorship (<http://www.icmje.org/recommendations/browse/roles-and-responsibilities/defining-the-role-of-authors-and-contributors.html>). There are in total 14 authors, and it is very possible that all should be listed as authors. However, with so many authors, I suggest you consider one more time if all authors fulfil the criteria for authorship. From the last sentence in the first paragraph on page 18, I understand that all authors fulfil the latter 3 out of 4 criteria listed for authorship. Regarding the first criteria, 13 out of 14 authors appear to fulfil this criteria (see table below). SJ is not mentioned in the part describing the first criteria. Was SW involved in other aspects than performing a search? What does it mean that OU and CN checked data, while other authors are described to have critically appraised studies? Please reconsider all 14 authors in terms of whether they should be listed as authors. Individuals who are not listed as authors may be mentioned under "Acknowledgements".

	Conception/Design	Acquisition	Analysis
RM	X	X	
WK	X	X	X
DG			X
HF		X	
LAK		X	
AA		X	
SW		X	
CN		X	
OU		X	
SJ			
AT		X	
CS	X	X	
STP	X		X
AC	X	X	X

Page 1: Title: Consider "mammographic screening" rather than "breast cancer screening", as you are interested in breast cancer screening with mammograms, not MRI, ultrasound, clinical examination, etc.

Abstract:

Objective: The focus of this study is factors associated with attendance and non-attendance, not the efficacy of mammography. I suggest omitting most of the first sentence, and replacing it with a sentence focusing on the latter part of the original sentence; that attendance varies. Both here and other places in the text you refer to "all factors". You will not be able to identify all factors that may potentially influence attendance. I suggest rephrasing to something along the lines of "all known" or "all identified".

Design: There is information regarding the end of the study period, but the start of the study period should also be mentioned.

Results: The highest OR was seen for country of origin, but is not mentioned here. See below regarding age and urban versus rural residence. I suggest including information regarding country of origin, and omitting age and urban residence.

Conclusion: This is a study including data from attendance back in time. I suggest rephrasing the first sentence to past tense. Also lower SES includes lower income. I would also rephrase to "Variations in service delivery, screening programmes and study populations...". Please explain (in response to editors and reviewers) in what way the last sentence has basis in the rest of the abstract, and if it doesn't, please consider deleting it.

Introduction:

First paragraph: In PROSPERO you refer to "worldwide predictors", but sentence 2 and 3 refer to UK data. Also, this review includes studies conducted outside the UK. It would be more relevant to refer to data on a more international level, even a European level would be more appropriate than UK only. In sentence 5 (and other places) you refer to a recent controversy. The controversy isn't recent, and reference 4 is from 2013 with data and discussion prior to 2013. Also, as far as I'm aware, reference 5 is not a reference that is relevant to this sentence.

Third paragraph: In the second sentence, you refer to all characteristics, but there are potentially unlimited factors that may influence attendance. I suggest specifying with "all identified" or "all available" or something similar. Further, this should not be considered as an update to Schueler's comprehensive review from 2008. Schueler's review excluded studies describing women who were not living in the United States, while this review includes studies from many countries.

Methods:

Search and Information sources:

First paragraph: Please specify when "...from inception..." is. Also, when you supply the search strategy in a supplementary file without the word limitations for the main text, I suggest providing the full search strategy.

Second paragraph: Please elaborate with some words what makes these individuals experts (e.g. experienced researchers with prior studies in the field).

Eligibility criteria: I take it that this means that the review could potentially include papers written in the period 1987 to June 2019. Please specify the period. Also, this is a study including papers from Europe, North America, Asia-Pacific and the Middle East. For studies conducted in other countries, the information regarding the Forrest report and inception of population-based screening in the UK is less relevant. Please rephrase this sentence in order to make it more relevant for the studies included (do you need to refer to the Forrest report? Was there population-based screening in other countries prior to 1987? Is the population-based screening relevant for the start of the study period considering that the review includes

settings with opportunistic screening?). I do not suggest altering the study period, but want to point out that when the review includes studies from many countries, information about the UK setting is not very relevant for readers from the Middle East, Scandinavia or the US. Could you simply write that the review included studies written between *month* 1987 and June 2019?

In the same paragraph I also question excluding studies with women in a narrow age range, only immigrants, only rural women and studies with number of invitations (with influence of repeated outcome) rather than number of women. You are interested in factors associated with attendance, and there may be relevant information also in studies with a selected group of women. Why is it relevant to include samples made up only of women who had previously attended (described on page 6), but not these groups? This point warrants a mention when discussing limitations.

Study selection and data extraction process:

First/Second paragraph: I understand from page 17/18 that the two reviewers assessing full-text studies and the two reviewers crosschecking were the same for all studies. Regarding titles and abstracts, please specify whether a reviewer was involved in screening all titles and abstracts, or if the five reviewers means that different reviewers reviewed different titles/abstracts, with the risk that some paper might have been excluded by some of the reviewers but not others.

Second paragraph: you specify "...who attended breast cancer screening and the number invited...". Does this mean that the study included both analyses of whether the women had ever/never attended, and whether they had attended for a specific period/screening round? If so, this should be specified, as there may be a difference between groups in the two categories. Further, please define the abbreviation SES (defined in the abstract, but not in the full text). Also, this would be a good place to define sociodemographic and socioeconomic clearly. You should for instance specify that while income is part of SES, results for income and SES are reported as two separate factors, as some studies report on income, while others on SES.

Synthesis of data: I miss an explanation for why a random effects model was chosen rather than for instance a fixed effects model. What is the limitation of choosing this model. You mention later that samples ranged from 82 individuals to almost 5 million. For the studies that were meta-analysed, what implications does the variation in population size have for the weighting. I suggest explaining why the model was chosen here, and elaborating more on the strengths and limitations of choosing this model in the discussion. Further, in the second paragraph you mention studies made up only of women who had previously attended screening. Further elaborating on this may clarify issues I've commented on other places in this feedback.

Results: In general, the results are described in a good and understandable way, and I thus only have minor comments.

Literature search: This would be an appropriate place to include the study period (start and end).

Study characteristics:

First paragraph: The implication of including studies with few

	participants should be addressed when discussing limitations and the model chosen. Third paragraph: Studies on self-reported mammograms were excluded due to risk of recall/reporting bias. It is likely that many of the studies had self-reported data regarding the nine factors listed. If so, this warrants a mention in the discussion. Further, some places in the text low numbers are written numerically. If indicated by the author guidelines, please write out low numbers with letters both here and other places in the text. Fourth paragraph: There is a reference to Supplementary file F. Please update the status for the study by Lagerlund in Supplementary file F. Quantitative data analysis (meta-analyses): Some results are reported three times, first in Table 1, then in the text, and then a third time in the figures. Please consider if it is necessary to report the same result three times. Is it possible to gather the figures on one page/in one large figure with sections for each forest plot, naming the different sections Figure 3a, 3b, 3c, etc, without the text becoming even smaller? In some of the figures, eg Figure 5, the text is already hard to read. Could the forest plots be moved to a supplementary file if you can't fit them on one page? Table 1: I suggest moving the cross after "Observational studies" in the box for age to a different place, as the text says "All results in this table...". There is also an end parenthesis missing at the end of the text under Table 1 (the parenthesis starting with OR is not closed, and there is a similar issue in the corresponding text, last sentence, third paragraph on page 10.). Also, please check the confidence intervals, I assume that the first confidence interval should be 0.88-1.08. Narrative studies: Table 2 gives a very good overview of studies that were not included in Table 1. Above there is a mention of urban versus rural in a separate paragraph. It is also mentioned separately in the discussion. The OR for three studies is 1.12 from 78,040 women. There are only two studies regarding access to vehicles, but these studies have 150,000 women and OR around 0.4. Why is then urban vs rural a point that is so important that it is included in the abstract (which I have suggested to delete) and discussion, compared to access to vehicles and other factors with clearly significant results here? Is a point more important simply if three studies investigate it and it can be meta-analysed, rather than if only two studies investigate it and it cannot, or do you have to take other factors into account as well (this point could also be explored in the discussion)? I suggest deleting the paragraph about urban residency on page 15 (single sentence paragraph). Another argument for downplaying the importance of urban vs rural is the finding from Le's (and mine) study in Table 2 for Oslo, which compares the capital city vs the rest of the country. Further, in Table 2, there are three studies regarding distance to screening centres. Why has this factor not been meta-analysed. Please revise the second paragraph. Immigration status does not belong to this section as it is explored under "Quantitative data
--	---

	analysis (meta-analyses)". In paragraph 4, I suggest rephrasing to say that they did not show any statistical difference, rather than writing that they "...appeared to have no association..." Discussion: Main findings: Please include country of origin as a main finding. Page 15, fourth paragraph: Please check the age groups in reference 29 and 30. Are the age groups and reference groups in these references relevant for the age groups in this review? I suggest deleting the part after the comma, and simply writing "We did not find a significant effect of age." without references. Page 15, fifth paragraph: I suggest deleting this sentence, or alternatively elaborate on the point. Strengths and limitations: First paragraph: Can you can say that you excluded studies that sampled population subgroups if you included studies with only women who had previously attended. In the third paragraph you should also specify that some factors will always be unknown. The potential number of factors influencing attendance is unlimited, and even with multivariate analyses, there would be factors unknown to us that would not be accounted for, and also factors of relevance that are hard to measure in quantitative studies and would not be included in a review that does not include qualitative studies (e.g. emotions, experiences and relations). A limitation that also should be discussed is the categories used. You, as authors, have to make some choices as you go along. Choosing the age range 50-59 vs 60-69 means that you might have gotten different results than if other age ranges were chosen. Similarly, some studies have shown that women with more years of higher education have lower attendance than women with less years of higher education. A discussion of choices made could be beneficial. It would also be appropriate to discuss the implications of your choice of focusing on all possible characteristics. Is the aim specific enough so that it is possible to answer it in a systematic review? Studies focusing on individual factors attempt to answer a single question, are you answering one question or several questions? What are the benefits of performing systematic reviews for individual factors, and what are the benefits of performing a systematic review with several factors as you do here? Conclusion: The conclusion should be re-written so that it is more to the point: First paragraph: I would argue that your main finding is that immigrants have lower attendance than non-immigrants, but this is not mentioned here (nor in the abstract). I'm also unsure if you should include the last sentence in this paragraph, what do you know regarding private screening facilities? Second paragraph: I struggle to see the relevance of informed choice in the conclusion in relation to the main findings reported in
--	---

	this study. Yes, informed choice is important in the setting of breast cancer screening, but do the results presented for instance in the abstract (which should be the most important results) imply that informed choice is important? Is the second paragraph a conclusion to this review?
--	---

REVIEWER	Dominika Bhatia University of Toronto, Institute of Health Policy, Management and Evaluation
REVIEW RETURNED	25-Apr-2021

GENERAL COMMENTS	Thank you for the opportunity to review this manuscript. The paper is well-written and addresses an important topic. The methodology is robust, with a prospectively-registered protocol and documented deviations from protocol. I have the following comments that may strengthen the manuscript:  1. In the Eligibility Criteria section (pg. 5), the authors state that studies were excluded if they reported “uptake data by number of invitations sent rather than the number of women” (8 studies excluded for this reason per Figure 1). In the Study Characteristics section (pg. 7), the authors mention that one of the reasons why 16 studies could not be pooled was because uptake data were reported by the “number of invitations sent rather than by number of patients”. What is the difference between the studies excluded for this reason and the studies kept in the review but not pooled for this reason? 2. In the Risk of Bias section (pg. 8), the authors state that “23% had a high RoB, mostly due to SES being measured at the area level (e.g., neighbourhood) rather than individual level.” Could the authors kindly elaborate on how the RoB was appraised for each study and why neighbourhood-level SES was deemed to present a higher RoB than individual-level SES? There is substantial evidence that the ecological environment plays an important and independent role in health status (e.g., due to access to health services, neighbourhood walkability, and availability of other resources). Therefore, while neighbourhood-level SES measures may capture a different underlying construct than individual-level wealth, education, or other SES components, they may not necessarily constitute a greater risk of bias from a methodological perspective (e.g., information bias due to misclassification). 3. As noted in the footnote to Table 1 (pg. 9), most of the studies included in the meta-analyses were observational. In light of significant heterogeneity, the possible correlation between the examined sociodemographic factors, and the possibility of residual confounding in meta-analyses of observational studies, it would be important to pool or highlight in the narrative synthesis the most adjusted estimates of effect size for each of the examined factors. In particular, the authors may consider highlighting the studies that have adjusted for covariates deemed to be critically important. 4. In Table 2 (pg. 11-14), it is a bit surprising that there were only 1-2 studies identified that examined the effect of comorbidity, depression, diabetes, and heart disease on breast cancer screening participation. Could the authors kindly elaborate on the efforts made to ensure comprehensiveness of the literature? For instance, the following systematic reviews have been published on these topics, each including 20-30 primary studies:  • Solmi et al (2020) Disparities in cancer screening in people with
---

	mental illness across the world versus the general population: prevalence and comparative meta-analysis including 4 717 839 people. Lancet Psychiatry. DOI: 10.2026/S2215-0366(19)30414-6  • Mitchell et al (2018) Breast cancer screening in women with mental illness: comparative meta-analysis of mammography uptake. The British Journal of Psychiatry. DOI: 10.1192/bjp.bp.114.147629 • Bhatia et al (2020) Breast, cervical and colorectal cancer screening in adults with diabetes: a systematic review and meta-analysis. Diabetologia. DOI: 10.1007/s00125-019-04995-7 • Diaz et al (2017) Association between comorbidity and participation in breast and cervical cancer screening: a systematic review and meta-analysis. Cancer Epidemiology. DOI: 10.1016/j.canep.2016.12.010 4. In the Discussion (pg. 17), the authors note that the studies “may also suffer from publication bias, which cannot be estimated with corroboratory evidence”. Was publication bias or presence of small study effects assessed using visual or statistical approaches in this review (e.g., funnel plots, Egger test)? 5. In Tables 1 and 2, it would be helpful to include the absolute screening uptake rates across the examined characteristics, as absolute differences may be more modest than the relative estimates. 6. In Supplementary File F, it appears that the “analysis status” and “explanation” are incomplete for the Lagerlund (2014) study.
--	--

VERSION 1 – AUTHOR RESPONSE

Reviewer 1		
4	Methods: What software was used to conduct the meta-analysis?	We have added a sentence at the end of the Methods section stating that all analyses were conducted in Stata v16.
5	1. The results are presented clearly although it would be helpful for the papers included in the meta analysis and narrative review to be referenced in the main body of the text rather than having to look at the supplementary material	We have cited all included studies in the main text and eliminated Supplemental file D
6	2. What scales were used to categorise women into low, medium and high income and SES groups?	All studies included in the SES analysis measured deprivation using a mix of the following area-based domains: income, employment, social or welfare support, housing, education, car access, crime rates, single parenthood, and variables related to foreign birth. These studies reported deprivation using scales of three or five categories, from least to most deprived. Studies using five categories were collapsed into three

		categories by combining attendance rates for categories two, three and four into one attendance rate to represent the medium deprivation category. The analysis of income included five studies that reported household income or total income with either three, four or five categories, from lowest to highest. For studies with four or five income categories, the middle categories were combined into a single attendance rate representing the intermediate income category.
7	3. Usual convention is to present p-values as $p < 0.001$ rather than $p = 0.000$ or $p < 0.000$	We have changed all p-values less than 0.001 to $p < 0.001$.
8	Table 1. It would be useful to have a column specifying the range of the ORs for each factor	We have added the range of ORs to Table 1.
9	Discussion: The review contains both RCT and observational study data. Have authors considered the influence of a given study design on the results in terms of potential biases?	The research question and design of this study precludes use of RCT data as RCT data per se. Instead, we extracted control arm population attendance rates from RCT data where possible. Thus, the whole study is observational in design and therefore carries with it those inherent biases. We have added a sentence to this effect in the discussion section
		in strengths and limitations
10	Figures 3-10: It would be helpful if the text size was larger	We have produced new versions of these figures and tried to make the text larger. However, another review comment (comment 36 in this table) asked if we could combine the figures onto one page or into one large figure. We have done this and also made the text larger and the image resolution larger.
11	Supplementary file E: It would be useful to have two additional columns here – the factor being analysed (e.g. SES, income, age etc.) and the odds ratios.	We have added these two columns to the table, along with 95% confidence intervals. It is now Supplementary file D

		(as per Comment 5 above).		
12	Supplementary file F: Lagerlund reference needs completing	We had originally intended to remove Lagerlund from Supplementary file F, and have now done so. Please note that this file is now Supplementary file E.		
Reviewer 2				
13	Page 1, 17 and 18: Please refer to the ICMJE criteria for authorship (http://www.icmje.org/recommendations/browse/roles-and-responsibilities/defining-the-role-of-authors-and-contributors.html). There are in total 14 authors, and it is very possible that all should be listed as authors. However, with so many authors, I suggest you consider one more time if all authors fulfil the criteria for authorship. From the last sentence in the first paragraph on page 18, I understand that all authors fulfil the latter 3 out of 4 criteria listed for authorship. Regarding the first criteria, 13 out of 14 authors appear to fulfil this criteria (see table below). SJ is not mentioned in the part describing the first criteria. Was SW involved in other aspects than performing a search? What does it mean that OU and CN checked data, while other authors are described to have critically appraised studies? Please reconsider all 14 authors in terms of whether they should be listed as authors. Individuals who are not listed as authors may be mentioned under "Acknowledgements".	We have cross checked and are comfortable that all authors contributed appropriately within the definitions of authorship provided by the journal. This study involved a large number of contributing studies and was originally part of a PhD. It was discussed and worked on over two year period by a team working together in the West Midlands CLAHRC Prevention and Detection theme. In particular OU and CN who are senior systematic reviewers undertook additional quality assessment of the data extraction process.		
		Conception/Design	Acquisition	Analysis
	RM	X	X	
	WK	X	X	X
	DG			X
	HF		X	
	LAK		X	
	AA		X	
	SW		X	
	CN		X	
	OU		X	
	SJ			
	AT		X	
	CS	X	X	
	STP	X		X
	AC	X	X	X
14	Page 1: Title: Consider "mammographic screening" rather than "breast cancer screening", as you are interested in breast cancer screening with mammograms, not MRI, ultrasound, clinical examination, etc.	Thank you for this suggestion. We have changed this in the abstract as well as ensuring that this is made clear in the main and supplementary		

		documents. We have kept the title as it stands because of its relevance for particular search strategies likely to be used in the future.
15	Abstract: Objective: The focus of this study is factors associated with attendance and non-attendance, not the efficacy of mammography. I suggest omitting most of the first sentence, and replacing it with a sentence focusing on the latter part of the original sentence; that attendance varies. Both here and other places in the text you refer to "all factors". You will not be able to identify all factors that may potentially influence attendance. I suggest rephrasing to something along the lines of "all known" or "all identified". Design: There is information regarding the end of the study period,	Thank you for these comments. We have modified the first sentence as suggested, and changed 'all factors' to 'all identified factors' throughout the document, including in the abstract and, for example, in the Strengths and Limitations list and on page 4 of the Introduction. We have also noted the beginning of the study period.
	but the start of the study period should also be mentioned.	
16	Abstract Results: The highest OR was seen for country of origin, but is not mentioned here. See below regarding age and urban versus rural residence. I suggest including information regarding country of origin, and omitting age and urban residence.	This is now included in the abstract.

17	Abstract Conclusion: This is a study including data from attendance back in time. I suggest rephrasing the first sentence to past tense. Also lower SES includes lower income. I would also rephrase to “Variations in service delivery, screening programmes and study populations...”. Please explain (in response to editors and reviewers) in what way the last sentence has basis in the rest of the abstract, and if it doesn’t, please consider deleting it.	We have changed the language to past tense, deleted ‘income’ and rephrased the next sentence as suggested. We have changed the last sentence to better reflect the content of the abstract.
18	Introduction: First paragraph: In PROSPERO you refer to “worldwide predictors”, but sentence 2 and 3 refer to UK data. Also, this review includes studies conducted outside the UK. It would be more relevant to refer to data on a more international level, even a European level would be more appropriate than UK only.	We have replaced the UK data with global and EU data from 2019 and 2020.
19	Introduction: In sentence 5 (and other places) you refer to a recent controversy. The controversy isn’t recent, and reference 4 is from 2013 with data and discussion prior to 2013. Also, as far as I’m aware, reference 5 is not a reference that is relevant to this sentence.	Thank you for these comments. We have removed the word ‘recently’ and removed the non-relevant reference.
20	Introduction: Third paragraph: In the second sentence, you refer to all characteristics, but there are potentially unlimited factors that may influence attendance. I suggest specifying with “all identified” or “all available” or something similar. Further, this should not be	We have indicated that we are addressing ‘all identified’ rather than ‘all’ characteristics. Thank you for pointing out that the Schueler review included only US studies. We have removed that reference.

	considered as an update to Schueler’s comprehensive review from 2008. Schueler’s review excluded studies describing women who were not living in the United States, while this review includes studies from many countries.	
21	Methods: Search and Information sources: First paragraph: Please specify when “...from inception...” is. Also, when you supply the search strategy in a supplementary file without the word limitations for the main text, I suggest providing the full search strategy.	While we searched most databases from inception to June 2019, we filtered results so as to retrieve only studies published since 1987. Therefore, we have modified the text to reflect the dates of study inclusion, which were January 1987 to June 2019. We have revised Supplementary file B so that it now includes full search strategies for all databases.
22	Second paragraph: Please elaborate with some words what makes these individuals experts (e.g. experienced researchers with prior studies in the field).	The text has been modified so it now includes the following: ‘Experienced researchers with prior studies in the field.’

23	Eligibility criteria: I take it that this means that the review could potentially include papers written in the period 1987 to June 2019. Please specify the period.	We have specified the period of study eligibility in the Methods section.
24	Eligibility criteria: Also, this is a study including papers from Europe, North America, Asia-Pacific and the Middle East. For studies conducted in other countries, the information regarding the Forrest report and inception of population-based screening in the UK is less relevant. Please rephrase this sentence in order to make it more relevant for the studies included (do you need to refer to the Forrest report? Was there population-based screening in other countries prior to 1987? Is the population-based screening relevant for the start of the study period considering that the review includes settings with opportunistic screening?). I do not suggest altering the study period, but want to point out that when the review includes studies from many countries, information about the UK setting is not very relevant for readers from the Middle East, Scandinavia or the US. Could you simply write that the review	Thank you for pointing out that the Forrest report is relevant to the UK but less relevant or necessary to explain the eligibility criteria. We have removed the reference to the Forrest report and done as suggested by stating the beginning and end dates of our search.
	included studies written between *month* 1987 and June 2019?	
25	Eligibility criteria: In the same paragraph I also question excluding studies with women in a narrow age range, only immigrants, only rural women and studies with number of invitations (with influence of repeated outcome) rather than number of women. You are interested in factors associated with attendance, and there may be relevant information also in studies with a selected group of women. Why is it relevant to include samples made up only of women who had previously attended (described on page 6), but not these groups? This point warrants a mention when discussing limitations.	We were strict in our eligibility criteria in including only studies from the perspective of population-based screening programmes. Studies where the sampling frame was restricted to certain demographic groups (and not based on population-based screening programmes) were

		excluded. This has been mentioned in the discussion.
26	Study selection and data extraction process: First/Second paragraph: I understand from page 17/18 that the two reviewers assessing full-text studies and the two reviewers crosschecking were the same for all studies. Regarding titles and abstracts, please specify whether a reviewer was involved in screening all titles and abstracts, or if the five reviewers means that different reviewers reviewed different titles/abstracts, with the risk that some paper might have been excluded by some of the reviewers but not others	Two individuals cross checked all screening for titles and abstracts.
27	Study selection and data extraction process: Second paragraph: you specify "...who attended breast cancer screening and the number invited...". Does this mean that the study included both analyses of whether the women had ever/never attended, and whether they had attended for a specific period/screening round? If so, this should be specified, as there may be a difference between groups in the two categories.	Most studies reported cross-sectional attendance data which included mixed groups of those who were attending for the first time and some who had previously attended. For some studies these two groups were separately identified, and we have reported this. We acknowledge that there may be a difference between these categories and have commented on this in the limitations section.

28	Further, please define the abbreviation SES (defined in the abstract, but not in the full text). Also, this would be a good place to define sociodemographic and socioeconomic clearly. You should for instance specify that while income is part of SES, results for income and SES are reported as two separate factors, as some studies report on income, while others on SES.	We have clarified that sociodemographic factors include socioeconomic status (SES), and as suggested, on page 4 we now clarify how SES was measured with the following text: “ ... socioeconomic status (SES, which was measured in two ways, a) with various composite indices of deprivation that included
		factors such as housing density, employment, education, social support, car ownership, and crime prevalence, and b) based on household income); ...”
29	Synthesis of data: I miss an explanation for why a random effects model was chosen rather than for instance a fixed effects model. What is the limitation of choosing this model. You mention later that samples ranged from 82 individuals to almost 5 million. For the studies that were meta-analysed, what implications does the variation in population size have for the weighting. I suggest explaining why the model was chosen here, and elaborating more on the strengths and limitations of choosing this model in the discussion.	We have added this text to the synthesis of data section: “Random effects models were used to allow for heterogeneity in the effects of the factors considered to vary across the various studies.” We also added a condensed version of the following to the Strengths and Limitations section: “We chose random effects models as almost all of our analyses contained heterogeneity and it is also expected that there would be differences in attendance across the different study populations. The random effects

		analysis relaxes the assumption that the same magnitude of effect is observed within each study, and so was the most suitable choice for this review. Studies with larger sample sizes are assumed to contain the least uncertainty and are given higher weightings than smaller studies. For analyses of small numbers of studies, the random effects analysis may struggle to correctly estimate uncertainty, but any meta-analysis performed on few studies would have its limitations, and the use of random effects analysis maintained consistency with the other analyses.” In addition to the above explanations, we also used metan and network forest commands in Stata, which both use DerSimonian and Laird random effects models. Ideally we would use Hartung-Knapp-Sidik-Jonkman adjustment however this has not yet been implemented within these Stata commands. However all methods have limitations when there are a small number of studies, or when the sample sizes vary considerably.
--	--	--

30	Synthesis of data: Further, in the second paragraph you mention studies made up	See comments to 25 and 27 above
	only of women who had previously attended screening. Further elaborating on this may clarify issues I've commented on other places in this feedback.	
31	Results: Literature search: This would be an appropriate place to include the study period (start and end).	I have added the start and end dates of our searches.
32	Study characteristics: First paragraph: The implication of including studies with few participants should be addressed when discussing limitations and the model chosen.	We have added an explanation of the implications of the model chosen in light of studies with few participants to the Strengths and Limitations section (also see comment 54 below).

33	Study characteristics: Thirds paragraph: Studies on self-reported mammograms were excluded due to risk of recall/reporting bias. It is likely that many of the studies had self-reported data regarding the nine factors listed. If so, this warrants a mention in the discussion.	We have mentioned the perspective of the study — population-based screening programmes — and that we have excluded self-reported attendance in the Methods section, under Eligibility, and in the Discussion, under Strengths and Limitations. We understand that for some of the factors (e.g. ethnicity, attitudes to screening) self-report is essential, so we have also noted this in the Discussion.
34	Study characteristics: Further, some places in the text low numbers are written numerically. If indicated by the author guidelines, please write out low numbers with letters both here and other places in the text.	We have made changes in line with the following author guideline: “Numbers under 10 are spelt out, except for measurements with a unit (8 mmol/l) or age (6 weeks old), or when in a list with other numbers.”
35	Fourth paragraph: There is a reference to Supplementary file F. Please update the status for the study by Lagerlund in Supplementary file F.	Thank you for pointing this out. Lagerlund 2014 has been removed from the table in Supplementary file F.

36	Quantitative data analysis (meta-analyses): Some results are reported three times, first in Table 1, then in the text, and then a third time in the figures. Please consider if it is necessary to report the same result three times. Is it possible to gather the figures on one page/in one large figure with sections for	Odds ratios and confidence intervals for meta-analyses are now reported Table 1 and the text. We acknowledge that there is overlap but are aware that people get their take-home messages differently and some only read tables and figures whereas other read text.
	each forest plot, naming the different sections Figure 3a, 3b, 3c, etc, without the text becoming even smaller? In some of the figures, eg Figure 5, the text is already hard to read. Could the forest plots be moved to a supplementary file if you can't fit them on one page?	We have combined the forest plots so that all single-comparison analyses (age, homeownership, marital status, etc) appear in one figure (Figure 3) and multi-comparison analyses (i.e. education, socioeconomic status, and income) appear in one figure (Figure 4). We have made the text as large as possible, but please advise if it is still not large enough and we will modify this further, for example, as you suggest, by making them supplementary files.
37	Table 1: I suggest moving the cross after "Observational studies" in the box for age to a different place, as the text says "All results in this table...". There is also an end parenthesis missing at the end of the text under Table 1 (the parenthesis starting with OR is not closed, and there is a similar issue in the corresponding text, last sentence, third paragraph on page 10.). Also, please check the confidence intervals, I assume that the first confidence interval should be 0.88-1.08.	We have moved the cross to the top of Table 1. Thank you for pointing out the missing end parentheses; we have modified this and hope it is now clearer. Indeed, the first confidence interval was incorrect and has now been corrected.

38	Narrative studies: Table 2 gives a very good overview of studies that were not included in Table 1. Above there is a mention of urban versus rural in a separate paragraph. It is also mentioned separately in the discussion. The OR for three studies is 1.12 from 78,040 women. There are only two studies regarding access to vehicles, but these studies have 150,000 women and OR around 0.4. Why is then urban vs rural a point that is so important that it is included in the abstract (which I have suggested to delete) and discussion, compared to access to vehicles and other factors with clearly significant results here? Is a point more important simply if three studies investigate it and it can be meta-analysed, rather than	Narrative studies: Thank you We agree that we may have overemphasised urban versus rural and that urban/rural differentials are no longer a good marker of access compared perhaps to vehicle (or public transport) availability. Urban versus rural has historically been an important distinguishing factor in terms of access - so that we felt it important to report the lack of difference that we found. We take your point about vehicle access but do not have space to include studies not meta-analysed in the abstract. We clearly need more studies on access to vehicles!
	if only two studies investigate it and it cannot, or do you have to take other factors into account as well (this point could also be explored in the discussion)? I suggest deleting the paragraph about urban residency on page 15 (single sentence paragraph). Another argument for downplaying the importance of urban vs rural is the finding from Le's (and mine) study in Table 2 for Oslo, which compares the capital city vs the rest of the country.	We have deleted the single sentence paragraph about rural/urban residence on page 15.
39	Further, in Table 2, there are three studies regarding distance to screening centres. Why has this factor not been metaanalysed.	The three studies reporting distance to screening centres measured distance using very different scales, which we did not feel could be meaningfully pooled for meta-analysis. Two studies measured the distance in kilometres, but using very different scales:  • Jensen 2012b: 0-

		20km / >20-40km / >40-60km / >60km)  • St-Jacques 2013: <2.5 km, 2.5-<5km / 5-<12.5km / 12.5-<25km / 25-<50km / 50-<75km >=75km Ouédraogo 2014, measured travel time in minutes: <=15 min / >15 min / missing. We mention this under Results/Narrative Synthesis (“Factors that could not be meta-analysed (because they were reported in fewer than three studies or could not be pooled) are reported in Table 2 with odds ratios”). However, we should have explained this in the Methods section as well. We have now modified the following sentence at the end of the Methods section, under Data Synthesis as follows: “We summarized results narratively if there were inadequate quantitative data for meta-analysis, if variables were reported in fewer than three studies,[17] or if the data from multiple studies were highly variable and therefore could not be meaningfully pooled.”
--	--	---

40	Please revise the second paragraph. Immigration status does not belong to this section as it is explored under “Quantitative data	Thank you for this comment. We have now removed the mention of immigration status in the Narrative synthesis section.
	analysis (meta-analyses)”.	
41	In paragraph 4, I suggest rephrasing to say that they did not show any statistical difference, rather than writing that they “...appeared to have no association...”	Thank you – we have rephrased this as suggested.
42	Discussion: Main findings: Please include country of origin as a main finding.	We have added this to the first sentence under Main Findings.

43	Page 15, fourth paragraph: Please check the age groups in reference 29 and 30. Are the age groups and reference groups in these references relevant for the age groups in this review? I suggest deleting the part after the comma, and simply writing “We did not find a significant effect of age.” without references. Page 15, fifth paragraph: I suggest deleting this sentence, or alternatively elaborate on the point.	We have made these changes.
44	Strengths and limitations: First paragraph: Can you can say that you excluded studies that sampled population subgroups if you included studies with only women who had previously attended.	We were clear throughout that our perspective was population screening programmes. In the study where we were therefore able to clearly identify that women had previously attended using routine data from population screening programmes we included this. Please also see response to comment 27.
45	Strengths and limitations: In the third paragraph you should also specify that some factors will always be unknown. The potential number of factors influencing attendance is unlimited, and even with multivariate analyses, there would be factors unknown to us that would not be accounted for, and also factors of relevance that are hard to measure in quantitative studies and would not be included in a review that does not include qualitative studies (e.g. emotions, experiences and relations).	Thank you for this suggestion. We have added text to this effect to the third paragraph.
46	Strengths and limitations: A limitation that also should be discussed is the categories used. You, as authors, have to make some choices as you go along. Choosing the age range 50-59 vs 60-69 means that you might have gotten different results than if other age ranges were chosen.	We have included a sentence to this effect in the strengths and limitations.

	Similarly, some studies have shown that women with more years of higher education have lower attendance than women with less years of higher education. A discussion of choices made could be beneficial.	
47	Strengths and limitations: It would also be appropriate to discuss the implications of your choice of focusing on all possible characteristics. Is the aim specific enough so that it is possible to answer it in a systematic review? Studies focusing on individual factors attempt to answer a single question, are you answering one question or several questions? What are the benefits of performing systematic reviews for individual factors, and what are the benefits of performing a systematic review with several factors as you do here?	Our research aim was to provide an overall systematic assessment of all identified factors affecting breast screening attendance from the perspective of population based screening programmes. Much research has been undertaken on varying different factors — but there is very little which systematically investigates all identified factors.
48	Conclusion: The conclusion should be re-written so that it is more to the point: First paragraph: I would argue that your main finding is that immigrants have lower attendance than non-immigrants, but this is not mentioned here (nor in the abstract).	Thank you for this comment. We have revised the conclusion to ensure harmonisation with the abstract and the main findings of the study and also based on other changes in the document as a result of the review process. It is now much shorter.
49	Conclusion: I'm also unsure if you should include the last sentence in this paragraph, what do you know regarding private screening facilities?	See comment to point 48

50	Second paragraph: I struggle to see the relevance of informed choice in the conclusion in relation to the main findings reported in this study. Yes, informed choice is important in the setting of breast cancer screening, but do the results presented for instance in the abstract (which should be the most important results) imply that informed choice is important? Is the second paragraph a conclusion to this review?	One of our underlying interests was to see if we could find an element of informed choice research in this review. There was none and it is clear that this comment in the conclusion is now out of place, so it has been removed.
Reviewer 3		
51	1. In the Eligibility Criteria section (pg. 5), the authors state that studies were excluded if they reported “uptake data by number of invitations sent rather than the number of women” (8 studies	Thank you for pointing this out. The sentence on page 7 states that there were 3 possible reasons that the 16 studies could not be analysed, one of which was that data were reported by number of
	excluded for this reason per Figure 1). In the Study Characteristics section (pg. 7), the authors mention that one of the reasons why 16 studies could not be pooled was because uptake data were reported by the “number of invitations sent rather than by number of patients”. What is the difference between the studies excluded for this reason and the studies kept in the review but not pooled for this reason?	invitations. In fact, only one study fell under this category (Romaine et al 2012). The other 15 studies could not be analysed or pooled because uptake data were reported by health-provider characteristics, because the paper reported percentage uptake but not sample sizes per category, or (we have added another reason in this revised draft, which applies to one study),

		because data for different factors were not reported separately. We had included Romaine 2012 in the review in error and have now removed it and updated Figure 1 (the PRISMA flow diagram) and other text and tables accordingly.
52	2. In the Risk of Bias section (pg. 8), the authors state that “23% had a high RoB, mostly due to SES being measured at the area level (e.g., neighbourhood) rather than individual level.” Could the authors kindly elaborate on how the RoB was appraised for each study and why neighbourhood-level SES was deemed to present a higher RoB than individual-level SES? There is substantial evidence that the ecological environment plays an important and independent role in health status (e.g., due to access to health services, neighbourhood walkability, and availability of other resources). Therefore, while neighbourhood-level SES measures may capture a different underlying construct than individual-level wealth, education, or other SES components, they may not necessarily constitute a greater risk of bias from a methodological perspective (e.g., information bias due to misclassification).	Thank you for this comment. How was risk of bias appraised? Risk of bias was assessed using the Quality in Prognosis Studies (QUIPS) tool. Each domain comprises several specific sources of bias, with guidance provided on how each item should be assessed. For example, within the ‘Prognostic Factors’ domain these are:  1. Definition of the prognostic factor (PF) 2. Valid and Reliable Measurement of PF 3. Method and Setting of PF Measurement 4. Proportion of data on PF available for analysis 5. Method used for missing data An example of the guidance notes (for item 2) is “Method of PF measurement is adequately valid and reliable to limit misclassification bias (e.g., may include relevant outside sources of information on measurement properties, also CS characteristics, such as blind measurement and limited reliance on

		recall).” Each of the specific biases is given a rating for (1) reporting (yes, partial, no, or unsure), and (2) risk of bias (high, moderate, low).
		Based on this information, an overall risk of bias rating is given for each domain. Domain level ratings were determined as follows: low risk = all items in domain rated as low risk; high risk = at least one item rated as high risk; moderate risk = any other combination. Two reviewers independently assessed each study, with final risk of bias ratings determined by consensus. Why was neighbourhood-level SES considered to be a high risk of bias compared to individual-level SES? Neighbourhood-level SES represents the average SES of individuals within a particular area. It may not reflect any individual and can be particularly influenced in areas with diverse communities. For example, a mixture of higher- and lower-level SES households could result in an area-level SES that is between the two, and accurately reflects neither. In our review, the unit of analysis was women who are eligible for

		screening (rather than people living within geographical areas). The majority of people (and, indeed, the average person in a neighbourhood) are not likely to be women who are eligible for breast cancer screening. Therefore, individual-level SES is more relevant (and less of a risk of bias) to our review than neighbourhood-level SES. We do agree that area-related factors, such as those you highlight, could be predictors of uptake. Indeed, we found some evidence relating to distance to screening centres. Though this was not significantly associated with uptake.
53	3. As noted in the footnote to Table 1 (pg. 9), most of the studies included in the meta-analyses were observational. In light of significant heterogeneity, the possible correlation between the examined sociodemographic factors, and the possibility of residual confounding in meta-analyses of observational studies, it would be important to pool or highlight in the narrative synthesis the most adjusted estimates of effect size for each of the examined factors. In particular, the authors may consider highlighting the studies that have adjusted for covariates deemed to be critically important.	This is a very fair point. Please also see our comments in response to comment 9 and the associated additions to the strengths and limitations section of the discussion. 'A second limitation is that we extracted univariable associations with uptake. In practice, many of the variables investigated will be highly correlated, and there will be complex interactions and confounding which we have not been able to account for. While some studies did report multivariable models, these were varied in

		structure, methods and variables included, so would have been difficult to combine in any meaningful way. We were therefore unable to undertake multivariable meta-regression analysis, examining the effects of individual attendance factors on overall attendance’.
54	4. In Table 2 (pg. 11-14), it is a bit surprising that there were only 1- 2 studies identified that examined the effect of comorbidity, depression, diabetes, and heart disease on breast cancer screening participation. Could the authors kindly elaborate on the efforts made to ensure comprehensiveness of the literature? For instance, the following systematic reviews have been published on these topics, each including 20-30 primary studies:  • Solmi et al (2020) Disparities in cancer screening in people with mental illness across the world versus the general population: prevalence and comparative meta-analysis including 4 717 839 people. Lancet Psychiatry. DOI: 10.2026/S2215-0366(19)30414-6 • Mitchell et al (2018) Breast cancer screening in women with mental illness: comparative meta-analysis of mammography uptake. The British Journal of Psychiatry. DOI: 10.1192/bjp.bp.114.147629 • Bhatia et al (2020) Breast, cervical and colorectal cancer screening in adults with diabetes: a systematic review and meta- analysis. Diabetologia. DOI: 10.1007/s00125-019-04995-7 • Diaz et al (2017) Association between comorbidity and participation in breast and cervical cancer screening: a systematic review and meta-analysis. Cancer Epidemiology. DOI: 10.1016/j.canep.2016.12.010 	Thank you for this comment. Indeed, there are many studies on comorbid conditions and breast cancer screening and, as you point out, several very recently published reviews on this topic. The two most recently published reviews you cite, by Solmi et al and Bhatia et al, included one study each that we agree should have been included in our review. We have now added these two studies (Chan et al 2014 and Chochinov et al 2009, which report on diabetes and schizophrenia, respectively). We also screened every study in the four reviews you cite to be sure we haven’t missed any others, and found no others that fit our criteria, so we are confident that our search strategy, while not perfect, is comprehensive in terms of locating the majority of relevant studies. In particular, most studies use self-reported mammographic screening attendance data, which we excluded from our review. Many other studies used a sub-

		sample of the population of women invited to attend population-wide screening programmes, rather than including all eligible women. For example, studies by Vigod et al 2011 and Guilcher et al 2014, use a subsample of women who responded to a health survey. Moreover, many of the studies looking at comorbidity are from the USA, where population-wide data are rarely available for mammographic screening, due to the fragmented nature of the health system.
55	4. In the Discussion (pg. 17), the authors note that the studies “may also suffer from publication bias, which cannot be estimated with corroboratory evidence”. Was publication bias or presence of small study effects assessed using visual or statistical approaches in this	We have now produced funnel plots for all meta-analyses and added them as Supplementary file G. We refer to them on page 18 of the Discussion and include the following text: To investigate the risk of reporting bias, we conducted funnel plots
	review (e.g., funnel plots, Egger test)?	(Supplementary file G), which demonstrated the high level of heterogeneity present between the studies in our analyses. Age was the only analysis where the studies disagree over the direction of attendance, however the disagreement is among larger studies, suggesting this is unlikely to be associated with biased reporting and instead down to the study heterogeneity. All other analyses, while having

		studies which disagree on the point estimate, have agreement as to which group is more or less likely to attend mammographic screening. Overall, we are not concerned about reporting bias.
56	5. In Tables 1 and 2, it would be helpful to include the absolute screening uptake rates across the examined characteristics, as absolute differences may be more modest than the relative estimates.	We have added percentage uptake for each factor (or variable) vs its reference category, in Tables 1 and 2.
57	6. In Supplementary File F, it appears that the “analysis status” and “explanation” are incomplete for the Lagerlund (2014) study.	We had originally intended to remove Lagerlund from this table, as its results are reported in Table 2 of the main document. We have now deleted the reference to Lagerlund 2014 in Supplementary file F (which is now Supplementary file E).